# Effects of Human Leukocyte Antigen DRB1 Genetic Polymorphism on Anti-Cyclic Citrullinated Peptide (ANTI-CCP) and Rheumatoid Factor (RF) Expression in Rheumatoid Arthritis (RA) Patients

**DOI:** 10.3390/ijms241512036

**Published:** 2023-07-27

**Authors:** Yu-Chia Chen, Chung-Ming Huang, Ting-Yuan Liu, Ning Wu, Chia-Jung Chan, Peng-Yu Shih, Hsin-Han Chen, Shih-Yin Chen, Fuu-Jen Tsai

**Affiliations:** 1Million-Person Precision Medicine Initiative, Department of Medical Research, China Medical University Hospital, Taichung 404, Taiwan; t92989@mail.cmuh.org.tw (Y.-C.C.); t92990@mail.cmuh.org.tw (T.-Y.L.); 2Division of Immunology and Rheumatology, Department of Internal Medicine, China Medical University Hospital, Taichung 404, Taiwan; hcm.jeffrey@msa.hinet.net; 3School of Chinese Medicine, China Medical University, Taichung 404, Taiwan; 4Department of Biological Sciences, Southeastern Oklahoma State University, Durant, OK 74701, USA; nwu@se.edu; 5Genetics Center, Department of Medical Research, China Medical University Hospital, Taichung 404, Taiwan; melody700525@yahoo.com.tw (C.-J.C.); u110001425@cmu.edu.tw (P.-Y.S.); 6Division of Plastic and Reconstructive Surgery, China Medical University Hospital, Taichung 404, Taiwan; scapulachenhh@yahoo.com.tw; 7Department of Medical Genetics, China Medical University Hospital, Taichung 404, Taiwan

**Keywords:** genome-wide association study (GWAS), phenome-wide association studies (PheWASs), rheumatoid arthritis, rheumatoid factor, anti-cyclic citrullinated peptide antibody, human leukocyte antigen DRB1

## Abstract

Rheumatoid arthritis (RA) is a systemic disease characterized by non-infectious inflammation of the joints and surrounding tissues, which can cause severe health problems, affect the patient’s daily life, and even cause death. RA can be clinically diagnosed by the occurrence of blood serological markers, rheumatoid factor (RF) and anti-cyclic citrullinated peptide antibody (anti-CCP). However, about 20% of RA patients exhibit negative results for both markers, which makes RA diagnosis difficult and, therefore, may delay the effective treatment. Previous studies found some evidence that human leukocyte antigen (HLA)-related genes might be the susceptibility genes for RA and their polymorphisms might contribute to varieties of susceptibility and disease severity. This study aimed for the genetic polymorphisms of the RA patient genome and their effects on the RA patient’s serological makers, RF and anti-CCP. A total of 4580 patients’ electronic medical records from 1992 to 2020 were retrieved from the China Medical University Hospital database. The most representative single-nucleotide polymorphisms (SNPs) were identified through a genome-wide association study (GWAS) followed by enzyme-linked immunosorbent assay (ELISA) validation using the blood from 30 additional RA patients. The results showed significant changes at the position of chromosome 6 with rs9270481 being the most significant locus, which indicated the location of the HLA-DRB1 gene. Further, patients with the CC genotype at this locus were more likely to exhibit negative results for RF and anti-CCP than those with the TT genotype. The C allele was also more likely to be associated with negative results for RF and anti-CCP. The results demonstrated that a genetic polymorphism at rs9270481 affected the expression of RF and anti-CCP in RA patients, which might indicate the necessity to develop a personalized treatment plan for each individual patient based on the genetic profile.

## 1. Introduction

Rheumatoid arthritis (RA) is a systemic disease characterized by non-infectious inflammation of the joints and surrounding tissues. RA is often accompanied by extra-articular manifestations (ExMs) and is known as a disease that can have systemic involvement. RA primarily affects small joints in the hands, feet, and wrists, and usually progresses slowly with temporary remissions. Due to multisystem involvement, autoantibodies can be detected in the blood serum, suggesting the autoimmune nature of the disease. The global prevalence of RA was approximately 0.24% in 2010 [1], while the rates in the United States and Nordic countries reached 0.5–1%, respectively [2,3], which reflected an annual incidence of about 40 cases per 100,000 population [2,4]. RA can affect individuals of all ages and genders with no preference on ethnic groups. However, the female-to-male ratio of incidence is approximately 3:1, and the age range is mostly between 25 and 60 years old [5,6,7]. The RA symptoms include synovitis, effusion, cell proliferation, granuloma formation, cartilage and bone tissue destruction, and eventually joint stiffness and functional impairment [8]. The main feature of RA is symmetric joint inflammation that can occur in all joints of the body, even resulting in ulceration. Patients may experience gradually reduced joint mobility and joint deformities, ultimately affecting the patient’s daily life and even causing death [9]. In most cases, RA developed progressively and could have long-term effects on the patient’s life including limb disability, functional loss, reduced life quality [10,11], and reduced work capacity [12]. The clinical diagnosis of RA is typically made by quantifying the disease characteristics of the patient, and different clinical domains contribute differently with different scores. According to the 2010 diagnosis criteria for rheumatoid arthritis published by the European League Against Rheumatism, RA is classified into four domains including joint involvement (score range of 0–5 points, based on the number of inflamed large/small joints), serology (score range of 0–3 points, based on negative/strong/weak positive RF and anti-CCP or ACPA), acute-phase reactants (score range of 0–1 points, based on normal/abnormal CRP and ESR), and symptom duration (2 levels; score range of 0–1 points, based on whether the duration is less or more than 6 weeks). The maximum score is set at 10 and the individual with a score of at least 6 is defined as the diagnosis of rheumatoid arthritis [13]. Previous studies have shown a high familial incidence rate of rheumatic diseases. The children whose parents had rheumatic diseases showed a higher risk of developing the disease than those without a family history. On the other hand, in monozygotic twins, the probability of one developing the disease was 20% if the other had the disease [14,15]. Some researchers have proposed a hypothesis that human leukocyte antigen (HLA)-related genes are susceptibility genes for RA, and their polymorphism also contributes to varieties of susceptibility and disease severity [16,17,18].

Rheumatoid factor (RF) and anti-cyclic citrullinated peptide antibody (anti-CCP) are important serological markers used to diagnose RA. Previous research has shown that RF demonstrated a sensitivity, specificity, and accuracy to the RA diagnosis of 91.7%, 74.4%, and 87.0%, respectively, while anti-CCP demonstrated a sensitivity, specificity, and accuracy of 88.0%, 90.4%, and 89.5%, respectively. The detection values after combining both markers were 90.2%, 83.3%, and 89.5%, indicating the importance of serological RF and anti-CCP results in the diagnosis of RA [19]. Our previous research has demonstrated that DNA repair genes, specifically MPG, OGG1, UNG, and EGFR, are RA susceptibility genes [20]. The polymorphisms of those identified genes including both SNPs and CNV contribute to the various susceptibilities of the disease. Moreover, those genes do not only enhance the risk of developing the disease but also impact the disease severity. However, it has been observed that around 20% of RA patients showed negative serological test results for both RF and anti-CCP with unknown mechanisms. Therefore, it is imperative to investigate the potential mechanisms and/or factors that may contribute to this type of phenomenon. This study focused on the genetic polymorphisms of RF and anti-CCP to investigate their potential impacts on the serological maker lab results in RA patients.

## 2. Results

### 2.1. Subjects and GWAS

A total of 4580 RA patients were initially included in this study. After the subject selection process, 857 patients were excluded due to the lack of RF and anti-CCP test data. Further grouping resulted in an experimental group of 1043 RA patients who were negative for both RF and anti-CCP and a control group of 805 patients who were positive for both RF and anti-CCP. GWAS was then conducted and resulted in 4019 SNPs that reached significant levels of *p* = 1 × 10^−5^ with SNPs on chromosome 6 even reaching the significance threshold of *p* = 5.0 × 10^−8^. The results of the Manhattan plot (Figure 1A) and QQ plot (Figure 1B) demonstrated that the significant locus was rs9270481 that belonged to the HLA-DRB1 gene (raw data was shown in Appendix A).

### 2.2. Genotype and Allele Frequencies of HLA-DRB1 rs9270481 SNP in RA Patients with/without RF and Anti-CCP Biomarkers

The statistical analysis of the association between the genotype and allele frequencies on rs9270481 and the RA patients’ blood serum tests showed that, at the rs9270481 locus, RA patients with the CC genotype had a higher probability of negative results in both serologic RF and anti-CCP tests than those with the TT genotype (OR = 4.55, *p* = 1.97 × 10^−23^) (Table 1). In terms of allele frequencies, patients with the C base were more likely to have negative results in both serologic RF and anti-CCP tests than those with the T base (OR = 1.96, *p* = 4.89 × 10^−23^). The results indicated that there was an association between the RA patient’s serum RF and anti-CCP levels and the rs9270481 locus on the HLA-DRB1 gene on chromosome 6. In addition, the genotype of the rs9270481 locus could affect the serum RF and anti-CCP results, while RA patients with the CC genotype or those carrying the C alkaline base were more likely to have negative results for both serum RF and anti-CCP (raw data was shown in Appendix A).

### 2.3. Analysis of Soluble HLA-DRB1 Molecules in the Serum of RA Patients

In this study, the levels of the HLA-DRB1 molecule (the transmembrane glycoprotein) were analyzed in 30 patients with clinically confirmed RA. The results showed that the levels of HLA-DRB1 were higher in patients of the TT genotype than those patients of the TC genotype with the lowest levels shown in patients of the CC genotype (Figure 2A). Similar results were observed when comparing the different genotypes within RA patients who were a T carrier of the rs9270481 SNP where the patients of the CC genotype demonstrated lower protein levels of HLA-DRB1 (Figure 2B). Although the resulting data did not show statistically significant differences, due to the fewer case numbers, HLA-DRB1 molecule levels still demonstrated the varieties among the different genotypes of RA patients (raw data was shown in Appendix A).

### 2.4. Correlation between Inflammatory Status and RA Patients with/without RF and Anti-CCP Biomarkers

ESR (erythrocyte sedimentation rate) and CRP (C-reactive protein) are longstanding laboratory tests that continue to be widely used in clinical practice. They represent the level of inflammation in the body at the time of the test. We collected the ESR and CRP data from the electronic medical records (EMRs) in China Medical University Hospital (CMUH) and analyzed the correlation between inflammatory status and RA markers. The results of the relationship between inflammatory status and RA patients with/without RF and anti-CCP biomarkers demonstrated that RA patients with RF and anti-CCP biomarkers, regardless of gender, exhibited a relatively severe inflammatory status. The results of statistical analysis showed very significant differences (*p* < 0.01) (Table 2) (raw data was shown in Appendix A). 

### 2.5. Biological Pathways and Functions Relevant to HLA-DRB1

Ingenuity Pathway Analysis (IPA) was applied to investigate the potential biological pathways involved or associated with HLA-DRB1. The human leukocyte antigen (HLA) region on chromosome 6 is the most crucial part of the human genome concerning rheumatoid arthritis (RA) pathogenesis. It contains genes that encode molecules responsible for regulating the immune response. The results demonstrated that HLA-DRB1 was implicated in numerous cellular pathways including apoptosis, cellular growth, proliferation and development, cellular immune response, pathogen-influenced signaling, and cytokine signaling, which suggested that HLA-DRB1 might play a crucial role in regulating and controlling cellular immune responses, and potentially serve as a key modulator of cell growth and apoptosis (Figure 3) (raw data was shown in Appendix A)

## 3. Discussion

Numerous GWAS have been conducted to investigate the relationship between RF and anti-CCP and RA. Okada et al. conducted a large-scale GWAS meta-analysis in a Japanese population, which identified 16 new RA susceptibility loci including well-known genes such as HLA-DRB1 and PTPN22 [21]. Eyre et al. conducted a GWAS of anti-CCP positive RA in a European population and identified several new susceptibility loci including the IL2RA gene [22]. Padyukov et al. conducted a GWAS of RA in a Swedish population and identified several new susceptibility loci including the CCL21 and CD40 genes [23]. Knevel et al. conducted a GWAS of RF and anti-CCP-positive RA in a Dutch population and identified several new susceptibility loci including the CCL21 and CD40 genes [24]. Orozco et al. conducted a GWAS of RA in a Spanish population and identified several new susceptibility loci including the CCR6 and CD226 genes [25].

Actually, a genetic association between RA and HLA variations was discovered early. Classical HLA haplotypes and their corresponding amino acid sequences have played a pivotal role in elucidating the functional implications of HLA associations with autoimmune diseases, including RA. The high linkage disequilibrium within HLA, specifically HLA-DRB1, contributes to the presence of multiple correlated SNPs associated with RA. Notably, the most significant disparity in the association of ACPA-positive RA between Asian and European populations lies in the amino acid at position 11 of HLA-DRB1, which confers the highest risk. Aspartic acid (Asp) is prevalent in the Asian population, whereas Valine (Val) predominates in the European population [26].

Collectively, these GWAS studies have confirmed the significant role of RF and anti-CCP in the development of RA, while also identifying novel genetic loci associated with RA. These findings provide valuable insights for exploring the pathogenesis and treatment of RA. This study conducted a GWAS of RA in a Taiwanese population and identified an association between the rs9270481 SNP of the HLA-DRB1 gene and the expression of anti-CCP and RF genes. And the position of the rs9270481 SNP is in chromosome 6: 32,591,268 base pairs (Chr6: 32,591,268), 2 KB upstream variant of HLA-DRB1 gene. The results suggested that RA individuals with the CC genotype exhibited a lower expression of RA markers. HLA-DRB1 is a member of the human leukocyte antigen (HLA) gene family and has been linked to the development of autoimmune diseases. Previous studies have identified several SNPs and HLA-DRB1 haplotypes associated with HLA-DRB1 [27,28]. However, the findings of rs9270481 SNP and its association with RA in this study are novel and have not yet been reported anywhere.

Both anti-CCP and RF are commonly used as the diagnostic markers for RA and the assessment indicator for the severity of disease [29]. The results of this study showed the lower expression levels of anti-CCP and RF markers in a Taiwanese population with the CC genotype, suggesting a lower risk for developing RA in such a genotype population compared to that of other genotypes. This finding may contribute to further understanding the pathogenesis of RA and provide more potential ways for developing effective RA treatments. It is important to note that GWAS analysis only identifies correlations between genes and diseases, rather than determining causal relationships. Therefore, further functional studies and clinical trials are needed to gain more comprehensive understandings of the pathogenic mechanisms and the clinical implications in the context of rheumatoid arthritis.

HLA molecules, classified into three classes, play various roles: class I and III present peptides from within the cell and activate the complement system, while class II is expressed on antigen-presenting cells and helps activate T-helper CD4+ cells by presenting peptides. Within HLA class II, variations in amino acid positions, particularly in the antigen-binding grooves of HLA-DR molecules, greatly contribute to the overall importance of the HLA region in RA [30]. Multiple pathways were identified to be associated with HLA-DRB1 by using Ingenuity Pathway Analysis (IPA) to analyze gene expression data. The biological pathways found in this study covered apoptosis, cellular growth, proliferation and development, cellular immune response, pathogen-influenced signaling, cytokine signaling, and others. HLA-DRB1 is an immune-related gene that encodes a subunit of the MHC-II molecule, which can bind to foreign antigens and present them to T cells, thus playing a crucial role in cellular immunity. Among these pathways found in this study, the cellular immune response and cytokine signaling were relevant to immunity, indicating that HLA-DRB1 might be involved in regulating and controlling cellular immune responses [31]. Further, the apoptosis, cellular growth, proliferation, and development pathways were related to cell apoptosis and growth. Therefore, HLA-DRB1 might also play a role in the growth and apoptosis of cells [32]. A pathogen-influenced signaling pathway was also found in this study, which implied that HLA-DRB1 might participate in signaling pathways relevant to pathogen infection [33]. This study identified the multiple pathways of HLA-DRB1, which were related to immunity, cell apoptosis and growth, and pathogen infection. The results suggested that further research on HLA-DRB1 might provide important insights into the regulation and control mechanisms of these pathways and have significant implications for related diseases.

The interpretation of our study results is limited because of the following: (1) Population specificity: Our study was conducted on a Taiwanese population, which may limit the generalizability of the findings to other populations with different genetic backgrounds. Further studies in diverse populations are needed to validate the results. (2) Sample size: Although our study initially included 4580 RA patients, after exclusions and grouping, the final sample size for the GWAS analysis was relatively small, with 1043 RA patients in the experimental group. A larger sample size would provide more statistical power and increase the reliability of the findings. (3) Causal relationship not established: Our study identified associations between genetic polymorphisms and serological markers, but we did not establish a causal relationship. Additional studies, including functional experiments and clinical trials, are necessary to determine the causal mechanisms underlying the observed associations. (4) Lack of functional validation: Our study focused on identifying genetic polymorphisms associated with RF and anti-CCP in RA patients but did not provide functional validation of the identified loci. Further experiments are needed to elucidate the functional implications of the identified genetic variants and their effects on RF and anti-CCP expression. (5) Limited exploration of other potential factors: Our study focused primarily on the genetic polymorphisms of HLA-DRB1 and their effects on RF and anti-CCP expression. Other factors, such as environmental factors and additional genetic variants, were not extensively explored. Future studies should consider a more comprehensive approach to investigate multiple factors involved in RA susceptibility and serological marker expression. (6) Lack of treatment response analysis: Our study mainly focused on the association between genetic polymorphisms and serological markers in RA patients. The impact of these genetic variants on treatment response or prognosis was not investigated. Assessing the relationship between genetic polymorphisms and treatment outcomes could provide valuable insights for personalized treatment strategies.

## 4. Material and Methods

### 4.1. Data Mining

We followed the 2010 diagnostic criteria from the European League Against Rheumatism, where RA is classified based on four domains: joint involvement (0–5 points), serology (0–3 points), acute-phase reactants (0–1 point), and symptom duration (0–1 point). The maximum score is 10, and a diagnosis of RA requires a score of at least 6 [13]. A total of 4580 clinically confirmed RA patients from China Medical University Hospital (CMUH) (Taichung, Taiwan, ROC) during the time period of 1992 to 2020 were involved in this study. The clinical information of all enrolled patients was retrieved from the electronic medical records (EMRs) of CMUH with the approval of CMUH ethics committees (approval numbers: CMUH111-REC1-176, CMUH107-REC3-058, and CMUH110-REC3-005).

### 4.2. Single-Nucleotide-Polymorphism (SNP) Data Processing

The TPMv1 customized SNP array (Thermo Fisher Scientific, Inc., Santa Clara, CA, USA) developed by the Academia Sinica and Taiwan Precision Medicine Initiative (TPMI) projects (Taipei, Taiwan, ROC) was employed in this study. A total of 714,457 SNPs were included for data analysis by using PLINK1.9 (https://www.cog-genomics.org/plink/1.9/ (accessed on 2 January 2023)) [34] with the subjects’ exclusion criteria based on high rates of missingness per marker (geno 0.1 > 10%) for SNPs and per individual (mind 0.1 > 10%) for subjects. Variants with a *p* value of Hardy–Weinberg equilibrium (hwe) < 10^−6^ (hwe 10^−6^) and minor allele frequency (MAF) < 10^−4^ (maf 0.0001) were also filtered out. After pre-screening data, a total of 4580 subjects with 508,004 variants remained in the study. Beagle 5.2 (https://faculty.washington.edu/browning/beagle/beagle.html (accessed on 9 January 2023)) was then applied for data imputation, and imputed data were further filtered by using an alternate allele dose <0.3 and genotype posterior probability < 0.9 [35]. After the data imputation and final filtration, a total of 4580 subjects with 9,607,262 variants were used for the GWAS analysis [36,37] (The flowchart of this study was shown in Appendix A and the raw data was shown in Appendix A).

### 4.3. Genome-Wide Association Study (GWAS)

PLINK 1.9 was employed in this study to generate the summary of statistics. The subjects were defined as the clinical cases that were recorded in the EMRs and diagnosed as rheumatoid arthritis. The data information of each individual case included patients’ blood test results of anti-CCP and RF. Based on the presence or absence of anti-CCP and RF, the subjects were divided into two groups. Logistic regression with multiple covariates, including sex and gender, was performed for data analysis, and the statistical significance was adjusted. Manhattan and quantile–quantile (QQ) plots with *p* values were generated by using R studio (https://posit.co/products/open-source/rstudio/ (accessed on 1 February 2023)) [38].

### 4.4. Determination of Soluble HLA-DRB1 Molecules in the Serum of RA Patients

An additional 30 clinically confirmed RA patients were recruited from CMUH. All patients were notified with the consent forms and the procedures were approved by the CMUH ethics committee. The patients were divided into 3 experimental groups based on their genotypes of the rs9270481 SNP with 10 patients in each group of TT, TC, and CC. The serological levels of the HLA-DRB1 molecule in RA patients were analyzed by using the Enzyme-linked immunosorbent assay (ELISA). Briefly, blood homogenates were centrifuged at 5000 rpm at room temperature for 5 min to collect the supernatants. The serum HLA-DRB1 molecule levels were then determined by using the Human HLA Class 2 Histocompatibility Antigen, DRB1 Beta Chain, and HLA-DRB1 ELISA kit (BT Lab, Shanghai, China) according to the manufacturer’s instructions [39,40]. ELISA was performed by using the Bio-Rad iMark™ Absorbance Microplate Reader (Bio-Rad Laboratories Inc., Hercules, CA, USA)

### 4.5. Explore the Relationship between Inflammatory Status and RA Patients with/without RF and Anti-CCP Biomarkers

To further investigate the relationship between inflammatory status and RA patients with/without RF and anti-CCP biomarkers, a retrospective study by using CMUH EMRs was performed. A total of 1802 clinically confirmed RA cases were involved in the study with 376 cases of male and 1426 cases of female in a male-to-female ratio of approximately 1:4. The lab results of major inflammatory indicators including erythrocyte sedimentation rate (ESR) and C-reactive protein (CRP) were reviewed and associated with the patients’ serum biomarker levels.

### 4.6. Analysis of Biological Pathway

The canonical pathway enriched by differential metabolites was analyzed by using the web-based QIAGEN Ingenuity Pathway Analysis (IPA) suite (http://www.ingenuity.com (accessed on 6 March 2023)) to identify relevant biological pathways and functions [41]. The analysis was performed by integrating a group of different metabolites retrieved from the Human Metabolome Database (HMDB), the false discovery rate (FDR) value, and the logarithmic fold change into the IPA. Enrichment pathways of different metabolites were then generated based on the Ingenuity Pathway Knowledge Database (QIAGEN, Hilden, Germany).

### 4.7. Statistical Analysis

Statistical analysis was performed by using SPSS (IBM, Armonk, New York, NY, USA) as described in our previous study [42]. Student’s *t*-test was used for the comparisons between two groups, while ANOVA was employed for the one-way analysis of variance among the groups. A *p* value less than 0.05 was defined as a significant difference, while a *p* value less than 0.01 was defined as a very significant difference.

## 5. Conclusions

In conclusion, our data suggested that HLA-DRB1 genetic polymorphism affects the expression of anti-cyclic citrullinated peptide (anti-CCP) and rheumatoid factor (RF) in rheumatoid arthritis (RA) patients. Further experiments are required to validate the results of this study and provide more comprehensive information to understand the potential mechanisms of HLA-DRB1 in RA development.

## Figures and Tables

**Figure 1 ijms-24-12036-f001:**
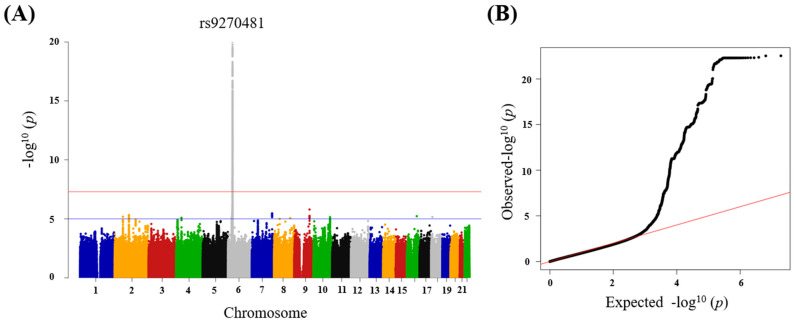
Association of genome-wide variants with rheumatoid arthritis (RA). (**A**) SNP on Manhattan plot. The upper and lower lines indicate the genome-wide significance threshold (*p* = 5.0 × 10^−8^) and the cut-off level for selecting SNPs (*p* = 1 × 10^−5^), respectively. (**B**) Observed *p* values versus expected one (the diagonal line) on QQ plot. The plot shows no significant deviation from the expected line, suggesting a lack of systematic biases or confounding effects in the analysis.

**Figure 2 ijms-24-12036-f002:**
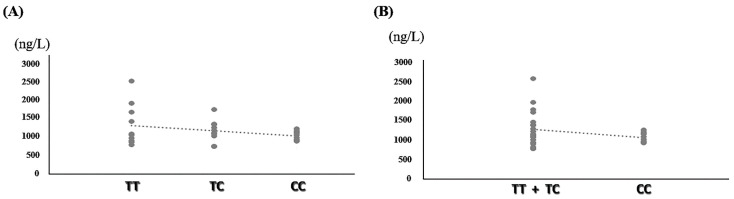
Analysis of HLA-DRB1 molecule levels in the serum of patients with rheumatoid arthritis (RA). (**A**) Compared the HLA-DRB1 molecule levels in RA patients with TT, TC, and CC genotype of rs9270481 SNP. (**B**) Compared the HLA-DRB1 molecule levels in RA patients with T carrier and CC genotype of rs9270481 SNP.

**Figure 3 ijms-24-12036-f003:**
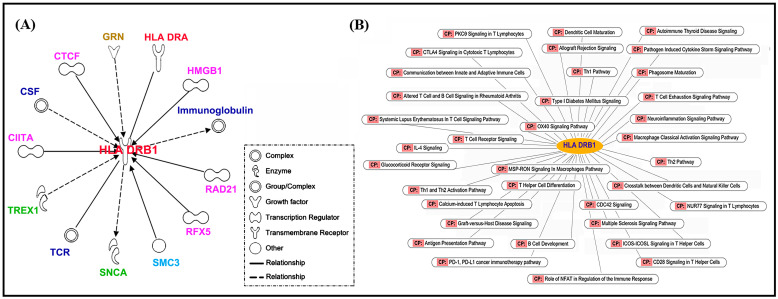
Ingenuity Pathway Analysis (IPA) to identify the biological pathways and functions that are relevant to HLA-DRB1. (**A**) The molecules interacting with HLA DRB1. (**B**) The signaling pathways in which HLA DRB1 is involved.

**Table 1 ijms-24-12036-t001:** Genotypic and allelic frequencies of rs9270481 genetic polymorphism in rheumatoid arthritis patients with/without RF/anti-CCP biomarkers.

dbSNP ID		RA Patient_RF/Anti-CCP_Positive	RA Patient_RF/Anti-CCP_Negative	OR (95% CI)	*p* Value
		(N = 805)	(N = 1043)		
rs9270481					
	Genotype	(N = 797)	(N = 1036)		
	CC	79 (9.9)	283 (27.3)	4.55 (3.37–6.15)	1.97 × 10^−23^
	CT	408 (51.2)	509 (49.1)	1.59 (1.28–1.96)	
	TT	310 (38.9)	244 (23.6)	Ref	
	Allele frequency				
	C	566 (35.5)	1075 (51.9)	1.96 (1.71–2.24)	4.89 × 10^−23^
	T	1028 (64.5)	997 (48.1)	Ref	

CI, confidence interval; OR, odds ratio.

**Table 2 ijms-24-12036-t002:** Correlation between inflammation status and RA patients with/without RF and anti-CCP biomarkers.

		Male + Female 1802 (100%)			Male 376 (20.87%)			Female 1426 (79.13%)		
		RF+/anti-CCP+	RF−/anti-CCP−	*p* Value	RF+/anti-CCP+	RF−/anti-CCP−	*p* Value	RF+/anti-CCP+	RF−/anti-CCP−	*p* Value
		795 (44.12)	1007 (55.88)		173 (46.01)	203 (53.99)		622 (43.62)	804 (56.38)	
ESR	Normal	390 (49.06)	700 (69.51)	<0.001	84 (48.55)	138 (67.98)	<0.001	306 (49.20)	562 (69.90)	<0.001
	Abnormal	405 (50.94)	307 (30.49)		89 (51.45)	65 (32.02)		316 (50.80)	242 (30.10)	
CRP	Normal	517 (65.03)	755 (74.98)	<0.001	84 (48.55)	131 (64.53)	<0.005	433 (69.61)	624 (77.61)	<0.001
	Abnormal	278 (34.97)	252 (25.02)		89 (51.45)	72 (35.47)		189 (30.39)	180 (22.39)	

## Data Availability

The original contributions presented in the study are publicly available. These data can be found at https://my.locuszoom.org/gwas/417572/?token=f797ae3dfda147dcb9779e786acec084 (accessed on 5 May 2023).

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
