# Peer review of "Effects of Human Leukocyte Antigen DRB1 Genetic Polymorphism on Anti-Cyclic Citrullinated Peptide (ANTI-CCP) and Rheumatoid Factor (RF) Expression in Rheumatoid Arthritis (RA) Patients"

_ijms, 2023, doi:10.3390/ijms241512036_

Round 1

Reviewer 1 Report

Suggestions for authors:

1. To change the structure of the article. Results an discussion should be after Methods.

2. To specify patients characteristics.

Author Response

Reviewer # 1 Comments:

  1. To change the structure of the article. Results and discussion should be after Methods.

Response:

We would like to thank the reviewer for the comment. We had re-checked the format online and confirmed its correctness of the structure of the article.

  1. To specify patients characteristics.

Response:

. We would like to thank the reviewer for the comment. We had added some description in the section of Materials and Methods (subchapter 4.1.) (page 11, line 268 to page 12, line 271 with blue color)

Reviewer 2 Report

The authors present a manuscript reporting that polymorphism of the gene coding for the HLA-DRB1 allele is associated with the presence or absence of antibodies to cyclic citrullinated protein and rheumatoid factor in patients with rheumatoid arthritis.

Remarks:

1)      Line 51: Replace "non-purulent" with "non-infectious"

2)      Line 53: The term "rheumatoid disease" is not generally used. Better to define that rheumatoid arthritis is a disease that can have systemic involvement.

3)      Line 70: It is incorrect to say that rheumatoid arthritis can be divided into the 4 categories according to the EULAR 2010 criteria. Different clinical domains contribute differently with different scores, and the achievement of at least (not more than) 6 points indicates the probability of diagnosis. Therefore, modify this section of the text in accordance with this clarification.

4)      Authors should specify which clinical parameters they used to define disease severity (DAS28, CDAI etc.).

5)      It is recommended to add a section on the limitations of the study

6)      Line 178: The statement that the presence of RF and/or anti-CCP correlates with the presence of autoantibodies is confusing, as RF and anti-CCP are the autoantibodies present in rheumatoid arthritis. Therefore, modify this statement

English should be improved

Author Response

Reviewer # 2 Comments:

The authors present a manuscript reporting that polymorphism of the gene coding for the HLA-DRB1 allele is associated with the presence or absence of antibodies to cyclic citrullinated protein and rheumatoid factor in patients with rheumatoid arthritis.

Remarks:

  1. Line 51: Replace "non-purulent" with "non-infectious"

Response:

We would like to thank the reviewer for the comment. We had revised the section of Abstract (page 2, line 36 with blue color) and Introduction (page 3, line 64 with blue color).

  1. Line 53: The term "rheumatoid disease" is not generally used. Better to define that rheumatoid arthritis is a disease that can have systemic involvement.

Response:

We would like to thank the reviewer for the comment. We had revised the section of Introduction (page 3, line 66 with blue color).

  1. Line 70: It is incorrect to say that rheumatoid arthritis can be divided into the 4 categories according to the EULAR 2010 criteria. Different clinical domains contribute differently with different scores, and the achievement of at least (not more than) 6 points indicates the probability of diagnosis. Therefore, modify this section of the text in accordance with this clarification.

Response:

We would like to thank the reviewer for the comment. We had revised the description in the section of Introduction (page 4, line 81 to 91 with blue color).

  1. Authors should specify which clinical parameters they used to define disease severity (DAS28, CDAI etc.).

Response:

We would like to thank the reviewer for the comment. We had added some description in the section of Discussion to show the limitation of this study (page 10, line 240 to page 11, 263 with blue color). In this study, our emphasis lies on investigating whether the genetic polymorphism of individuals with rheumatoid arthritis (RA) affects the expression of rheumatoid factor (RF) and cyclic citrullinated peptide (CCP). We appreciate the valuable suggestion from the reviewer, and in the future, we can further analyze the association between the expression of RF and CCP with disease activity score 28 (DAS28), Clinical Disease Activity Index (CDAI), and other relevant factors.

  1. It is recommended to add a section on the limitations of the study.

Response:

We would like to thank the reviewer for the comment. We had added some description in the section of Discussion (page 10, line 240 to page 11, 263 with blue color).

  1. Line 178: The statement that the presence of RF and/or anti-CCP correlates with the presence of autoantibodies is confusing, as RF and anti-CCP are the autoantibodies present in rheumatoid arthritis. Therefore, modify this statement.

Response:

We would like to thank the reviewer for the comment. We had revised the description in the section of Discussion (page 9, line 208 to 209 with blue color).

  1. Comments on the Quality of English Language: English should be improved.

Response:

We would like to thank the reviewer for the comment. We have re-edited the English by co-author Dr. Wu, whom the professor in Southeastern Oklahoma State University, USA.

Reviewer 3 Report

This study aimed on the genetic polymorphisms of RA patient genome and their effects on RA patient’s serological makers, RF and anti-CCP. Totally, 1,043 RA patients negative for both RF and anti-CCP, and 805 patients positive for both RF and anti-CCP were analyzed. Authors used Single Nucleotide Polymorphism (SNP) data processing, Genome-Wide Association Study (GWAS) and Enzyme-linked immunosorbent assay (ELISA).

The article deals with Asian population. Since HLA association with RA in Asian, European and African populations is not completely identical, I have found this paper interesting, but I have some important recommendation for improving the article.

 Major points:

1.     In general, authors used very modern technology and sophisticated statistical analyses to prove very old data from seventies of the last century. HLA DRB1 association with RA is known for almost 50 years. This fact should appear in the Introduction together with the chronology of development of knowledge about it. Since the history of HLA predisposition to RA is long, it should be reflected also in the Discussion that should be much longer.

2.     On another hand, in the References, there are citied many old articles (about 10 years old), and just few up-today articles. You should mention more new (e.g.: Padyukov L. Genetics of rheumatoid arthritis, 2022; Dedmon LE. The genetics of rheumatoid arthritis, 2020).

3.     You have found that RA patients with the CC genotype had a higher probability of negative results in both RF and anti-CCP tests than those with the TT genotype. Which HLA DRB1 alleles include C base, which HLA DRB1 alleles include T base, and where in the gene these bases are located? This information is important from the functional point of view, because certain regions of the gene code for certain epitopes of the protein.

4.     In the Results (subchapter 2.4.), you mentioned that RA patients with RF and anti-CCP exhibited a relatively severe inflammatory status, but you measured only ESR and CRP. I think you have not shown enough data for this statement.

5.     The detail function of HLA in immune system is very well described. You should mention it more correctly in the Results (subchapter 2.5.) and the Discussion (the third paragraph).

Minor points:

You should check again your English and eliminate many technical and grammar mistakes.

1.    The Results, subchapter 2.3. Analysis of serum HLA-DRB1 protein levels of RA patients: the HLA molecules are transmembrane glycoproteins (not proteins)!!! It is possible to detect their soluble form in the serum, but then it should be mentioned. Change the title into “Analysis of soluble HLA-DRB molecules in the serum of RA patients” Remove the word “protein” (line 135 and 137).

2.    Similarly, the Materials and Methods, subchapter 4.4. Change the title into “Determination of soluble HLA-DRB molecules in the serum of RA patients”, and remove the word “protein” (line 250). On the line 247, you have a big mistake: “adiponectin” should be changed into “HLA-DRB”.

3.    The Table 1: Correct statement “RA Patient_RF / anti-CCP_negative (not nositive)”.

4.    The Table 2: You have a big mistake on the last line: Correct the number “2520“ into the number “252“ (RF- / antiCCP- with abnormal CRP).

I recommend this paper for acceptation after major revision in the journal.

Moderate editing of English language required.

Author Response

Reviewer # 3 Comments:

This study aimed on the genetic polymorphisms of RA patient genome and their effects on RA patient’s serological makers, RF and anti-CCP. Totally, 1,043 RA patients negative for both RF and anti-CCP, and 805 patients positive for both RF and anti-CCP were analyzed. Authors used Single Nucleotide Polymorphism (SNP) data processing, Genome-Wide Association Study (GWAS) and Enzyme-linked immunosorbent assay (ELISA).

The article deals with Asian population. Since HLA association with RA in Asian, European and African populations is not completely identical, I have found this paper interesting, but I have some important recommendation for improving the article.

Major points:

  1. In general, authors used very modern technology and sophisticated statistical analyses to prove very old data from seventies of the last century. HLA DRB1 association with RA is known for almost 50 years. This fact should appear in the Introduction together with the chronology of development of knowledge about it. Since the history of HLA predisposition to RA is long, it should be reflected also in the Discussion that should be much longer.

Response:

We would like to thank the reviewer for the comment. We had added some description in the section of Discussion (page 8, line 187 to 195 with blue color).

  1. On another hand, in the References, there are citied many old articles (about 10 years old), and just few up-today articles. You should mention more new (e.g.: Padyukov L. Genetics of rheumatoid arthritis, 2022; Dedmon LE. The genetics of rheumatoid arthritis, 2020).

Response:

We would like to thank the reviewer for the comment. We had added the references [26] and [28] in the manuscript (page 8, line 195 and page 9, line 206 with blue color). We also revised the order of references.

[26].   Padyukov, L. (2022) Genetics of rheumatoid arthritis. Semin. Immunopathol. 44, 47-62.

[28].   Dedmon, L.E. (2020). The genetics of rheumatoid arthritis. Rheumatology (Oxford). 59, 2661-2670.

  1. You have found that RA patients with the CC genotype had a higher probability of negative results in both RF and anti-CCP tests than those with the TT genotype. Which HLA DRB1 alleles include C base, which HLA DRB1 alleles include T base, and where in the gene these bases are located? This information is important from the functional point of view, because certain regions of the gene code for certain epitopes of the protein.

Response:

We would like to thank the reviewer for the comment. We had some description in the section of Discussion (page 9, line 201 to 202 with blue color).

  1. In the Results (subchapter 2.4.), you mentioned that RA patients with RF and anti-CCP exhibited a relatively severe inflammatory status, but you measured only ESR and CRP. I think you have not shown enough data for this statement.

Response:

We would like to thank the reviewer for the comment. We had added some description in the section of Results (page 7, line 154 to 158 with blue color).

  1. The detail function of HLA in immune system is very well described. You should mention it more correctly in the Results (subchapter 2.5.) and the Discussion (the third paragraph).

Response:

We would like to thank the reviewer for the comment. We had added some description in the section of Results (subchapter 2.5.) (page 7, line 166 to 169 with blue color). And we also added description and a reference in the section of Discussion (the third paragraph) (page 9, line 218 to page 10, line 223 with blue color).

[30]. Wysocki, T., Olesińska, M., Paradowska-Gorycka, A. (2020) Current Understanding of an Emerging Role of HLA-DRB1 Gene in Rheumatoid Arthritis-From Research to Clinical Practice. Cells. 9, 1127.

Minor points:

  1. You should check again your English and eliminate many technical and grammar mistakes.

Response:

We would like to thank the reviewer for the comment. We have re-edited the English by co-author Dr. Wu, whom the professor in Southeastern Oklahoma State University, USA.

  1. The Results, subchapter 2.3. Analysis of serum HLA-DRB1 protein levels of RA patients: the HLA molecules are transmembrane glycoproteins (not proteins)!!! It is possible to detect their soluble form in the serum, but then it should be mentioned. Change the title into “Analysis of soluble HLA-DRB molecules in the serum of RA patients” Remove the word “protein” (line 135 and 137).

Response:

We would like to thank the reviewer for the comment. We had added some description and changed the title in the section of Results (subchapter 2.3) (page 6, line 141 to line 143 with blue color).

  1. Similarly, the Materials and Methods, subchapter 4.4. Change the title into “Determination of soluble HLA-DRB molecules in the serum of RA patients”, and remove the word “protein” (line 250). On the line 247, you have a big mistake: “adiponectin” should be changed into “HLA-DRB”.

Response:

We would like to thank the reviewer for the comment. We had changed the title in the section of Materials and Methods (subchapter 4.4.) (page 13, line 303 with blue color) and revised the mistakes (page 13, line 308 and line 311 with blue color).

  1. The Table 1: Correct statement “RA Patient_RF / anti-CCP_negative (not nositive)”.

Response:

We would like to thank the reviewer for the comment. We had revised the mistakes in Table 1.

  1. The Table 2: You have a big mistake on the last line: Correct the number “2520“ into the number “252“ (RF- / antiCCP- with abnormal CRP).

Response:

We would like to thank the reviewer for the comment. We had revised the mistakes in Table 2.

  1. I recommend this paper for acceptation after major revision in the journal.

Response:

We are very grateful for the support from the reviewer and thank you very much for everything.

Reviewer 4 Report

The work, in its synthesis, shows very clearly hypotheses, data, conclusions, limits and future goals. The supplementary material provided is extensive and more than satisfactory, demonstrating beyond the synthesis and brevity of the work, the enormous amount of data and studies carried out. 

Introduction provide sufficient background and include all relevant references, all the cited references  are relevant to the research,  the research design is appropriate, all  methods are adequately described. The results are clearly presented and the conclusions supported the results. English is good.

Author Response

Reviewer # 4 Comments:

  1. The work, in its synthesis, shows very clearly hypotheses, data, conclusions, limits and future goals. The supplementary material provided is extensive and more than satisfactory, demonstrating beyond the synthesis and brevity of the work, the enormous amount of data and studies carried out.

Introduction provide sufficient background and include all relevant references, all the cited references  are relevant to the research,  the research design is appropriate, all  methods are adequately described. The results are clearly presented and the conclusions supported the results. English is good..

Response:

We are very grateful for the support from the reviewer and thank you very much for everything.

Round 2

Reviewer 2 Report

The authors responded thoroughly to my inquiries.

English needs minor changes.

Reviewer 3 Report

Authors have answered all my questions and comments.